# Modeling the Reaction Process for the Synthesis of Ethyl Chrysanthemate from Ethyl Diazoacetate in a Micro-Flow Platform

**DOI:** 10.3390/mi16020125

**Published:** 2025-01-22

**Authors:** Dawei Xin, Yangcheng Lu

**Affiliations:** State Key Laboratory of Chemical Engineering, Department of Chemical Engineering, Tsinghua University, Beijing 100084, China; xindawei@huaan.com.cn

**Keywords:** cyclopropanation process, ethyl chrysanthemate, ethyl diazoacetate, reaction mechanism, kinetic model

## Abstract

Ethyl diazoacetate can react with 2,5-dimethyl-2,4-hexadiene to yield ethyl chrysanthemumate, an important raw material for synthesizing various pesticides. In conventional conditions, this cyclopropanation process suffers from low efficiency and yield due to ethyl diazoacetate. This demands more understanding of the catalytic process from the mechanism and modeling to find a solution. In this work, we set up a micro-flow platform to carefully study the kinetic characteristics of the cyclopropanation reaction of ethyl diazoacetate catalyzed by a complex of copper stearate and phenylhydrazine. Through a reasonable simplification of the reaction network, we established a reaction kinetic model with good prediction capacity within a wide range of operating conditions. It provides a basis for guiding the development of efficient conversion processes and condition optimization.

## 1. Introduction

Ethyl 2,2-dimethyl-3-(2-methylprop-1-enyl)cyclopropane carboxylic acid, also known as ethyl chrysanthemate, is the simplest-structured pyrethroid pesticide and is also an important intermediate for the synthesis of pyrethroid pesticides such as allethrin, permethrin, and prallethrin. In addition, chlorine and bromine substituted compounds of ethyl chrysanthemate are also important raw materials for the synthesis of many common pyrethroid pesticides such as deltamethrin, cypermethrin, and bifenthrin [1].

There are mainly two synthesis routes for ethyl chrysanthemate, namely the Martel route and the dimethylhexadiene route [2], as shown in Figure 1. Comparatively, the dimethylhexadiene process has the advantages of a short production process, relatively abundant raw material sources, and low cost, and is the most widely adopted synthesis process of ethyl chrysanthemate in the industry. In this process, ethyl diazoacetate reacts with olefin 2,5-dimethyl-2,4-hexadiene to undergo a cyclopropanation reaction under catalytic conditions. Ethyl diazoacetate is extremely unstable and is prone to releasing nitrogen to generate diethyl fumarate, diethyl maleate, and various other by-products, and even accidents such as explosions occur [3,4]. In order to ensure safety and deal with the instability of ethyl diazoacetate, for the batch-stirred reactor used in the industry, the addition of ethyl diazoacetate is carried out in a dripping manner, the apparent temperature is controlled at 20~80 °C, and the reaction time is as long as several hours, but there still exists the problem of low yield from ethyl diazoacetate to ethyl chrysanthemate (about 0.7). In order to develop a more efficient ethyl chrysanthemate production technology, it is also necessary to start from research on reaction mechanisms and kinetic modeling as fundamentals for the reaction environment and process regulation.

Regarding the cyclopropanation reaction of ethyl diazoacetate with various olefins catalyzed by metal ligand catalysts, in the past decades, researchers have continuously put forward models to explain the reaction mechanism [5,6,7,8,9,10]. The main reaction steps proposed in the catalytic cyclopropanation process are summarized in Figure 2. In detail, when the metal ligand catalyst (1) participates in the reaction, it can combine with ethyl diazoacetate (2) to form a complex (3), and then release one molecule of N_2_ to form a metal carbene (4); the metal carbene (4) can react with the olefin (5) to form the main product of the three-membered ring (6) and then detach from the catalyst (7), or react with another molecule of ethyl diazoacetate to form a dimer (8), or react with the C-C double bond in the dimer to generate a trimer (9); the metal ligand catalyst and the Lewis base undergo a coordination equilibrium to form a complex (10). In addition to the metal carbene, there is also a free carbene (11) in the system, and the free carbene also generates the main product of the three-membered ring, a dimer, and a trimer through reaction paths similar to those of the metal carbene. This complicated reaction network means that it is difficult to accurately predict the reaction kinetics or construct a simple kinetic model directly.

As for controlling the reaction environment and process, the microreactor is an effective platform due to excellent heat transfer [11,12,13,14], mixing [15,16,17], and safety performance [18,19,20]. There have been many successful cases of improving the reaction process efficiency and yield [21,22,23,24,25], especially suitable for reaction systems involving unstable intermediates and dangerous chemical processes [26,27,28]. In this work, we set up a micro-flow platform to carefully study the kinetic characteristics of the catalytic cyclopropanation reaction of ethyl diazoacetate, and were devoted to establishing a quantitative kinetic model based on a simplified reaction network which can predict efficiency and yield well. It provides a basis for both the process condition optimization and the system innovation for ethyl chrysanthemate synthesis and other ethyl diazoacetate-participating reactions.

## 2. Experimental Section

The main reagents included ethyl diazoacetate (C_4_H_6_O_2_N_2_, >91.0%, Aladdin, Bejing, China), 1,2-dichloroethane (C_2_H_4_Cl_2_, Xilong Chemical, Shantou, China), 2,5-dimethyl-2,4-hexadiene (C_8_H_14_, >97.0%, Aladdin, China), diethyl fumarate (C_8_H_12_O_4_, 99%, KarMar, Shanghai, China), diethyl maleate (C_8_H_12_O_4_, 99%, Sinopharm, Beijing, China), dimethyl phthalate (C_10_H_10_O_4_, 99.5%, Guangfu, China), and ethyl chrysanthemate (C_12_H_20_O_2_, 95%, J&K, Beijing, China). All these reagents are of analytical purity grade and were used without further treatment. Among them, 1,2-dichloroethane was used as the reaction solvent; ethyl diazoacetate and 2,5-dimethyl-2,4-hexadiene were reactants, denoted as EDA and DMH, respectively; both diethyl fumarate and diethyl maleate were the dimers of EDA, denoted as DIM; ethyl chrysanthemate was the product, denoted as EC; and dimethyl phthalate was the internal standard for gas chromatography analysis. The catalyst was a complex of copper stearate and phenylhydrazine, commonly used in the industry. GC data were collected on Agilent-7890A with an HP-INOWAX capillary column (30 m, 0.32 mm i.d.) and TCD detector.

The micro-flow platform is shown in Figure 1. The ethyl diazoacetate solution containing a catalyst and the 2,5-Dimethyl-2,4-hexadiene solution were separately delivered into stainless steel preheating tubes (o. d. 1.6 mm, i. d. 0.9 mm, length 1.0 m) by two peristaltic pumps. And then, the two feeds were mixed in a T-junction mixer with its outlet connecting the stainless steel reaction pipelines (including 6 sections of 2 m in length, of which 4 sections were 1 mm in inner diameter, denoted as S1, and 2 sections were 2.0 mm in inner diameter, denoted as S2). Through a seven-way ball valve, one section (S1) or more sections (at most four S1 and two S2) could be connected into the system, corresponding to the volume of the reaction pipeline adjustable from 1.7 mL to 20.4 mL (accessory volumes included). The connection method is shown in Figure 1b. The preheating tube and the reaction pipeline were both placed in a thermostat (Jinhua, China, ±0.1 K) to ensure temperature control. At the end of the reaction pipeline, a cooling tube (o. d. 3 mm, i. d. 2 mm, length 2 m) was connected to cool the reaction solution in an ice water bath, and a back-pressure valve was connected behind the cooling tube to ensure 0.7 to 0.8 MPa (gauge) back pressure to avoid evaporation in reaction pipelines.

## 3. Results and Discussion

### 3.1. Kinetic Characteristics of the Cyclopropanation Reaction

In this work, all the experiments were carried out with an excessive addition of 2,5-dimethyl-2,4-hexadiene over ethyl diazoacetate, repeated three times to provide average results and error analysis; the flow velocity in the reaction tube was around 0.1 m/s; the Reynold number was between 200 and 500, corresponding to a laminar flow; and the Peclet number was between 20 and 50, corresponding to little back-mixing. Firstly, fixing the initial EDA concentration at 0.02 mol/L, catalyst concentration at 0.02 mmol/L, and temperature at 130 °C, respectively, comparative experiments with different olefin excess amounts were carried out, and the results are shown in Figure 2. The EDA conversion reflected the consumption rate of ethyl diazoacetate from feeding to outlet. The DIM yield reflected the ratio of the generated dimers (the sum of diethyl fumarate and diethyl maleate) to the initial ethyl diazoacetate. The EC yield reflected the ratio of the generated ethyl chrysanthemate to the initial ethyl diazoacetate. As can be seen, the cyclopropanation reaction of EDA and DMH can basically achieve complete conversion within 1 min. During the reaction conversion process, the yield of the main product ethyl chrysanthemate gradually increases, the yield of the by-product dimers can reach a high level at around 10 s, and then a competition between dimer generation and consumption is presented by the reaction with the metal carbene, resulting in a plateau followed by a slow decrease.

Then, fixing the catalyst concentration at 0.02 mmol/L, the temperature at 130 °C, and the ratio of initial EDA concentration to DMH concentration at 1:1.1, respectively, comparative experiments with different reactant concentrations were carried out, and the experimental results are shown in Figure 3. It can be seen that as the concentrations of DMH and EDA were increased simultaneously, the yields of the main product ethyl chrysanthemate and the by-product dimers did not change significantly, remaining around 0.2 and 0.3, respectively. At the same time, with the increase in reactant concentrations, the reaction rate was significantly increased.

Furthermore, fixing the initial EDA concentration at 0.02 mol/L, the DMH concentration at 0.022 mmol/L, and the catalyst concentration at 0.02 mmol/L, respectively, the influence of the reaction temperature was investigated, and the results are shown in Figure 4. It can be seen that with the increase in the reaction temperature, the reaction rate was significantly increased, and at the same time, the dimer yield slightly increased, which implies that the activation energy of the reaction step for dimer generation was higher than the activation energy of the reaction step for dimer consumption.

### 3.2. Kinetic Analysis and Modeling

Considering the complexity of the reaction network shown in Figure 2, we attempted to ignore the mechanism steps that had no obvious influence on the apparent kinetics, simplified the kinetic modeling process, and thereby realized an acceptable prediction of the reaction process and achieved a plausible design of the intensified cyclopropanation reaction process. Herein, we considered the following assumptions: (1) the free carbene reaction pathway was negligible because the amount of products generated via the free carbene within the same time was significantly lower than that via the metal carbene; (2) the reaction steps for generating the polymer were negligible; (3) the differences in the reaction characteristics of the cis–trans isomers were negligible; (4) the reaction intermediates other than the metal carbene were negligible. Thus, we obtained the simplified reaction network shown in Figure 3.

This simplified reaction network included four reaction steps and one coordination equilibrium:

Step 1$$EDA+\mathrm{Catalyst}\overset{\;\;\;k_{0}\;\;\;}{\to }N_{2}+\mathrm{MC}$$

Step 2$$\mathrm{MC}+\mathrm{DMH}\overset{\;\;\;k_{1}\;\;\;}{\to }\mathrm{Catalyst}+\mathrm{EC}$$

Step 3$$\mathrm{MC}+\mathrm{EDA}\overset{\;\;k_{2}\;\;\;}{\to }\mathrm{Catalyst}+\mathrm{DIM}$$

Step 4$$\mathrm{MC}+\mathrm{DIM}\overset{\;\;\;k_{3}\;\;\;}{\to }\mathrm{Catalyst}+\mathrm{trimer}$$

Step 5$$\mathrm{Catalyst}+DMH\overset{\;\;K\;\;\;}{↔}\mathrm{Cat}\text{-}\mathrm{complex}$$
where MC corresponds to the metal carbene, and Cat-complex is the complex of the catalyst and DMH. According to these steps, we obtained five equations:(1)$$\frac{dC_{\mathrm{EDA}}}{dt}=-k_{0}C_{\mathrm{EDA}}C_{\mathrm{Cat}\text{-}\mathrm{complex}}-k_{2}C_{\mathrm{MC}}C_{\mathrm{EDA}}$$(2)$$\frac{dC_{\mathrm{MC}}}{dt}=k_{0}C_{\mathrm{EDA}}C_{\mathrm{Cat}\text{-}\mathrm{complex}}-k_{1}C_{\mathrm{MC}}C_{\mathrm{DMH}}-k_{2}C_{\mathrm{MC}}C_{\mathrm{EDA}}-k_{3}C_{\mathrm{MC}}C_{\mathrm{DIM}}$$(3)$$\frac{dC_{\mathrm{EC}}}{dt}=-\frac{dC_{\mathrm{DMH}}}{dt}=k_{1}C_{\mathrm{MC}}C_{\mathrm{DMH}}$$(4)$$\frac{dC_{\mathrm{DIM}}}{dt}=k_{2}C_{\mathrm{MC}}C_{\mathrm{EDA}}-k_{3}C_{\mathrm{MC}}C_{\mathrm{DIM}}$$(5)$$C_{\mathrm{cat}}=\frac{C_{\mathrm{cat},0}-C_{\mathrm{MC}}}{1+KC_{\mathrm{DMH}}}$$

Then, based on the experimental results under different initial reactant concentrations, olefin excess ratios, and temperatures, supposing that *K* is insensitive to temperature, the rate constants *k*_i_ of each conversion step in the kinetic model were regressed, and the results are shown in Table 1 as follows.

It can be seen that under various temperatures, *k*_1_ < *k*_2_ and *k*_3_ is always true, indicating that the competitive ability of the reaction substrate DMH on the metal carbene intermediate is weaker than that of EDA and the dimer. This means that in the subsequent design of process intensification, theoretically, to achieve a high reaction yield, a reaction environment with an excess of olefin should be introduced.

The calibrated curves obtained using the above parameters agreed with the experimental values well, as shown in Figure 2, Figure 3 and Figure 4. According to the reaction rate constants at different temperatures, from the Arrhenius equation, there is a linear relationship between ln*k* and 1/*T*. By plotting ln*k* − 1/*T*, the slope −*E*_a_/*R* and the intercept ln*A* can be obtained, thereby determining the reaction activation energy and the pre-exponential factor, as shown in Figure 5. The detailed activation energy and the pre-exponential factor of the reaction steps are listed in Table 2.

At this point, the kinetic model of the copper stearate–phenylhydrazine-catalyzed EDA cyclopropanation reaction has been obtained:(6)$$k_{1}=1.4989\times 10^{13}e^{-\frac{71.77\;\mathrm{kJ}/\mathrm{mol}}{RT}}s^{-1}\cdot (\mathrm{mol}/L{)}^{-1}$$(7)$$k_{2}=8.0898\times 10^{17}e^{-\frac{104.07\;\mathrm{kJ}/\mathrm{mol}}{RT}}s^{-1}\cdot (\mathrm{mol}/L{)}^{-1}$$(8)$$k_{3}=5.3287\times 10^{14}e^{-\frac{78.49\;\mathrm{kJ}/\mathrm{mol}}{RT}}s^{-1}\cdot (\mathrm{mol}/L{)}^{-1}$$(9)$$k_{0}=1.9555\times 10^{15}e^{-\frac{89.60\;\mathrm{kJ}/\mathrm{mol}}{RT}}s^{-1}\cdot (\mathrm{mol}/L{)}^{-1}$$

According to the simplified reaction network, after the metal ligand catalyst forms the metal carbene (corresponding to the rate constant *k*_0_) with EDA, there are three directions of transformation that the metal carbene will undergo: reacting with DMH to form the main product through ring formation (*k*_1_), reacting with EDA to form the dimer by-product (*k*_2_), and reacting with the dimer to form the trimer by-product through ring formation (*k*_3_). Judging from the regression results, the reaction activation energy of the main reaction step is the lowest. The low activation energy means that with the increase in the reaction temperature, the selectivity of the main reaction will show a gradually decreasing trend. As a result, if it is desired to utilize the excellent heat transfer performance, safety, and controllability of the microreactor to realize an efficient flow reaction process under high-temperature conditions, a relatively high olefin excess ratio should be introduced in the process design towards higher yield.

As the temperature decreases, the reaction selectivity can be improved, but the required reaction time increases significantly. Therefore, both the reaction yield and the reaction time are the objects that we need to pay attention to when screening the process conditions. And based on the kinetic model established previously, we can calculate the reaction yield under different temperatures, initial reactant concentrations, and reaction times, thereby guiding the screening of the process conditions.

### 3.3. Reaction Kinetics Prediction Based on the Kinetic Model

Figure 6 shows the reaction yields under different temperatures and reaction time conditions calculated by the kinetic model. The region of the purple rectangle in the figure shows the measurement condition range for model parameter regression; that is, the kinetic experimental data were obtained within temperatures from 110 °C to 140 °C and a residence time of less than 1 min. It is first necessary to test the accuracy of the model, especially for a wider temperature and time condition range than in the kinetic experiments to observe whether our kinetic model based on the simplified reaction network is suitable for extrapolation.

The calculation results show that at a lower reaction temperature, after a longer reaction time, as shown in the upper right part of Figure 6, the cyclopropanation reaction can achieve a higher reaction yield. In order to test the prediction ability of the kinetic model, we designed semi-batch stirring experiments at temperatures of 60 and 70 °C, and measured the reaction yield after the temperature-controlled reaction for 3 h. Under these two temperature conditions, the experimentally obtained reaction yields are 0.516 and 0.494, respectively, and the corresponding model-calculated yields are 0.543 and 0.497, respectively. Therefore, we verified that the reaction kinetic model can achieve relatively accurate prediction of the experimental results in a wider range of operating conditions, reflecting the plausibility of reaction network simplification.

## 4. Conclusions

Ethyl diazoacetate can react with 2,5-dimethyl-2,4-hexadiene to yield ethyl chrysanthemumate, an important raw material for synthesizing various pesticides. In conventional conditions, this cyclopropanation process of ethyl diazoacetate suffers from a long process time, low utilization of olefins, and so on. In order to develop a more efficient ethyl chrysanthemate production technology, it is necessary to start from the research on reaction mechanisms, reaction kinetic modeling, and regulation of the reaction environment and process. In this work, we set up a micro-flow platform, systematically studied the kinetic characteristics of the cyclopropanation reaction of ethyl diazoacetate catalyzed by copper stearate–phenylhydrazine, and explored a reasonable simplification of the complex reaction network capable of process predication or design. Through omission of the free carbene pathway in the ethyl diazoacetate catalytic cyclopropanation reaction, the by-product of polymers above trimer, the reaction difference of cis–trans isomers, and the reaction pathway of intermediates other than the metal carbene, we obtained a kinetic model of the ethyl diazoacetate catalytic cyclopropanation reaction as well as four reaction steps and the coordination equilibrium relationship of 2,5-dimethyl-2,4-hexadiene competing for the ligand of the metal catalyst. This kinetic model can well describe the reaction results under the copper stearate–phenylhydrazine catalytic system, and achieve accurate prediction of the reaction results within the expanded temperature, concentration, and time ranges as well. It provides a basis for guiding the development of efficient conversion processes and condition optimization for ethyl diazoacetate catalytic cyclopropanation and similar reactions.

## Data Availability

The data presented in this study are available on request from the corresponding author.

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
