# Peer review of "Modeling the Reaction Process for the Synthesis of Ethyl Chrysanthemate from Ethyl Diazoacetate in a Micro-Flow Platform"

_micromachines, 2025, doi:10.3390/mi16020125_

Round 1

Reviewer 1 Report

Comments and Suggestions for Authors

The manuscript discussed the effect of various thermodynamic and kinetic attributes of the reaction of ethyl diazoacetate with 2,5-dimethyl-2,4-hexadiene to generate ethyl chrysanthemate by using a new micro reaction experimental device. The manuscript could be accepted for publication after the following changes/additions:

1. What is the rational of incorporating this new micro reaction set-up? What is its advantage and what is background of the design of this instrument?

2. The formula of EDA in Figure 1b contains the 3H-diazirine. This structure would not be a stable form. It would be appropriate to use "N2=CHCO2Et".

3. In Figure 3, please give some detailed explanations on the set-up and operation procedures.

Author Response

Comment 1: What is the rational of incorporating this new micro reaction set-up? What is its advantage and what is background of the design of this instrument?

Response 1:

The use of a microreactor can rapidly initiate a reaction through the quick mixing of reactants and promptly quench the reaction by rapidly cooling the reaction system. In this way, by simply controlling the volume from the mixing point to the cooling point and the volumetric flow rate of the reactants, the residence time of the reaction can be strictly controlled. On the premise of fixing the volumetric flow rate of the reactants, we use different series combinations of multiple delay tubes to change the residence time of the reaction. The series combination can be changed through a seven-way valve without shutting down the system. These practices can provide strictly controllable reaction conditions and an accurate and seamlessly switchable residence time sequence, thereby improving the efficiency of experiments. This is also the background and advantage of the device design.

Comment 2: The formula of EDA in Figure 1b contains the 3H-diazirine. This structure would not be a stable form. It would be appropriate to use "N2=CHCO2Et".

Response 2:

Thank you for the reviewer's comments. The structural formula of EDA has been revised.

Comment 3: In Figure 3, please give some detailed explanations on the set-up and operation procedures.

Response 3:

Thank you for the reviewer's comments. Details of the experimental setup and operation method have been added.

Reviewer 2 Report

Comments and Suggestions for Authors

Review of the paper

 Modeling the reaction process for the synthesis of ethyl chrysanthemate from ethyl diazoacetate in a micro-flow platform

A micro-flow platform to carefully study the kinetic characteristics of the cyclopropanation reaction of ethyl diazoacetate catalyzed by the complex of copper stearate and phenylhydrazine was used in the paper.

The paper is fairly written in general and deserves to be published.

There are however a few points to be fixed:

1) Fig. 1 and comments in the text.

Please provide the detailed information (maybe in a special Table) on the length and volume of each combination of sections allowing to switch the volume of the reaction pipeline from 1.7 mL to 20.4 mL.

The corresponding ways of flow should be shown as well.

1a) The ice water bath should be shown in Fig. 1 as well.

2) The reagents used appear to be fully soluble one in the other. Hence, the homogeneous flow has been formed in pipelines (unlike many other studies with water-organic heterogeneous mixtures).

This fact should be specified in the Experimental Section in an explicit form.

3) Flow rates of each pump P1 and P2 should be provided in Experimental Section. Besides, specific information characterizing the flow regime should be also provided, like mean velocity in microchannels, Reynolds number, Peclet number.

All this data should demonstrate advantages of the microreactor process compared to the batch process.

Author Response

Comment 1: Fig. 1 and comments in the text.

Please provide the detailed information (maybe in a special Table) on the length and volume of each combination of sections allowing to switch the volume of the reaction pipeline from 1.7 mL to 20.4 mL.

The corresponding ways of flow should be shown as well.

1a) The ice water bath should be shown in Fig. 1 as well.

Response 1:

Thank you for the reviewer's comments. The length and volume of each part of the reaction device have been described. The flow pattern has also been explained. An ice-water bath is shown in Figure 1.

Comment 2: The reagents used appear to be fully soluble one in the other. Hence, the homogeneous flow has been formed in pipelines (unlike many other studies with water-organic heterogeneous mixtures).

This fact should be specified in the Experimental Section in an explicit form.

Response 2:

Thank you for the reviewer's comments. The reaction system is homogeneous, which has been stated in the experimental section.

Comment 3: Flow rates of each pump P1 and P2 should be provided in Experimental Section. Besides, specific information characterizing the flow regime should be also provided, like mean velocity in microchannels, Reynolds number, Peclet number.

All this data should demonstrate advantages of the microreactor process compared to the batch process.

Response 3:

Thank you for the reviewer's comments. The flow rates of the two pumps have been described in the experimental section. The ranges of the average flow velocity, Reynolds number, and Peclet number in the channel have been calculated and explained. The flow regime is laminar. In this paper, the advantage of the microreactor process over the batch process lies in the ability to strictly control the reaction environment and process, thus obtaining accurate reaction kinetic data. This is the basis for understanding the mechanism and accurately designing the reaction process.